# CRISPR/Cas9-mediated somatic correction of a novel coagulator factor IX gene mutation ameliorates hemophilia in mouse

Yuting Guan[1,†], Yanlin Ma[2,3,†,*], Qi Li[2], Zhenliang Sun[4], Lie Ma[1], Lijuan Wu[1], Liren Wang[1], Li Zeng[1], Yanjiao Shao[1], Yuting Chen[1], Ning Ma[2], Wenqing Lu[1], Kewen Hu[1], Honghui Han[5], Yanhong Yu[3], Yuanhua Huang[2], Mingyao Liu[1,6,**] & Dali Li[1,***]

## Abstract

The X-linked genetic bleeding disorder caused by deficiency of coagulator factor IX, hemophilia B, is a disease ideally suited for gene therapy with genome editing technology. Here, we identify a family with hemophilia B carrying a novel mutation, Y371D, in the human *F9* gene. The CRISPR/Cas9 system was used to generate distinct genetically modified mouse models and confirmed that the novel Y371D mutation resulted in a more severe hemophilia B phenotype than the previously identified Y371S mutation. To develop therapeutic strategies targeting this mutation, we subsequently compared naked DNA constructs versus adenoviral vectors to deliver Cas9 components targeting the *F9* Y371D mutation in adult mice. After treatment, hemophilia B mice receiving naked DNA constructs exhibited correction of over 0.56% of *F9* alleles in hepatocytes, which was sufficient to restore hemostasis. In contrast, the adenoviral delivery system resulted in a higher corrective efficiency but no therapeutic effects due to severe hepatic toxicity. Our studies suggest that CRISPR/Cas9-mediated *in situ* genome editing could be a feasible therapeutic strategy for human hereditary diseases, although an efficient and clinically relevant delivery system is required for further clinical studies.

**Keywords** gene therapy; genome editing; hemophilia B; hemostasis; monogenetic disease

**Subject Categories** Genetics, Gene Therapy & Genetic Disease; Haematology

See also: **TH Nguyen & I Anegon** (May 2016)

## Introduction

Hemophilia B (HB), an X-linked genetic bleeding disorder caused by deficiency of coagulator factor IX (FIX), affects 1 of every 25,000 to 30,000 males worldwide (Thompson & Chen, 1993). Based on the FIX plasma procoagulant levels, the disease is classified as mild (5–40% of normal activity), moderate (1–5% of normal activity), and severe (< 1% of normal activity; White *et al*, 2001). As solely increasing the plasma FIX levels as low as 1% results in significant restoration of clotting activity, HB is considered a good model for evaluating the efficacy of distinct gene therapy strategies. Introducing the *F9* gene cDNA into the liver (the natural source of FIX secretion) of HB animal models and patients through recombinant adeno-associated virus (rAAV) has proven to be efficacious (Kay *et al*, 1993, 2000; Snyder *et al*, 1997; Nathwani *et al*, 2014), but two major concerns remain: the duration of expression and the safety issue of AAV-mediated random insertion of the transgene into the host genome. Sleeping Beauty transposon-induced random insertion of the FIX transgene in HB mice has shown that integration into the host genome is feasible for long-term and high-level FIX expression (Yant *et al*, 2000). Recent studies demonstrated that site-specific integration of human *F9* cDNA into transcriptionally highly active genomic loci in hepatocytes through either a nuclease (zinc finger nuclease) dependent or independent manner successfully ameliorated the disease in mice (Li *et al*, 2011; Anguela *et al*, 2013; Barzel *et al*, 2015). These strategies theoretically overcome the significant concerns of transgene expression duration and random insertion-induced safety issue, but whether direct correction of the *F9* mutation by targeting the endogenous locus is sufficient for restoration of clotting activity through somatic genome editing is still not determined.

1 Shanghai Key Laboratory of Regulatory Biology, Institute of Biomedical Sciences and School of Life Sciences, East China Normal University, Shanghai, China
2 Hainan Provincial Key Laboratory for Human Reproductive Medicine and Genetic Research, Hainan Reproductive Medical Center, the Affiliated Hospital of Hainan Medical University, Hainan Medical University, Haikou, China
3 Department of Obstetrics and Gynecology, Nanfang Hospital, Southern Medical University, Guangzhou, China
4 Fengxian Hospital affiliated to Southern Medical University, Shanghai, China
5 Bioray Laboratories Inc., Shanghai, China
6 Department of Molecular and Cellular Medicine, The Institute of Biosciences and Technology, Texas A&M University Health Science Center, Houston, TX, USA
*Corresponding author. Tel: +86 898 66776091; E-mail: mayl1990@foxmail.com
**Corresponding author. Tel: +86 021 54345014; E-mail: myliu@bio.ecnu.edu.cn
***Corresponding author. Tel: +86 021 24206824; E-mail: dlli@bio.ecnu.edu.cn
†These authors contributed equally to this work

The CRISPR/Cas9 system developed from an RNA-mediated adaptive immune system identified in bacteria is a revolutionary technology for gene editing in cells and organisms (Jinek *et al*, 2012; Cong *et al*, 2013; Hwang *et al*, 2013; Li *et al*, 2013; Mali *et al*, 2013; Shen *et al*, 2013; Wang *et al*, 2013; Doudna & Charpentier, 2014; Niu *et al*, 2014). Cas9/sgRNA induces site-specific DNA double-strand breaks (DSBs) which then initiate either error-prone non-homologous end joining (NHEJ) or homology-directed repair (HDR) in the presence of donor DNA templates (Cong *et al*, 2013; Mali *et al*, 2013). Pioneer studies have shown potential applications of the CRISPR/Cas9 system for correction of genetic disorders in cells or mouse embryos (Schwank *et al*, 2013; Wu *et al*, 2013; Long *et al*, 2014), as well as prevention of cardiovascular disease in adult mice through disruption of the *PCSK9* gene (Ding *et al*, 2014). Recently, a mouse model of hereditary tyrosinemia type I(HT1)caused by a point mutation of fumarylacetoacetate hydrolase (FAH) was phenotypically restored via Cas9-mediated gene repair *in vivo* (Yin *et al*, 2014). In this particular model, the repaired cells had a survival advantage and expanded to replace the mutant hepatocytes when the pharmacological inhibitor NTBC was withdrawn (Yin *et al*, 2014). As the initial correction rate in the model is lower to 0.4%, it is essential to investigate the efficacy of Cas9-mediated *in vivo* genetic correction of other heritable diseases in which the repaired cells cannot be easily selected for repopulation.

Here, we identify a family with HB carrying a novel FIX mutation which is confirmed as an HB causative mutation through generation of mice with an identical mutation via the CRISPR/Cas9 system. Delivery of the Cas9 component *in vivo* resulted in correction of over 0.56% of endogenous *F9* alleles in hepatocytes and restored hemostasis in mice. Our data strongly demonstrate that correction of genetic disorders through repair of mutations *in situ* via CRISPR/Cas9-mediated genome editing is feasible.

# Results

### Identification of novel *F9* mutation in HB patients

A 9-year-old male proband (Fig 1A IV:2) was diagnosed HB with an abnormal activated partial thromboplastin time (aPTT) of 84 s (reference values: 25 s ~35 s) with a normal factor VIII activity. His clotting activity was remarkably decreased to about 2% of the normal level (Table 1). By taking a detailed family history, five patients were identified in the pedigree (Fig 1A). A single missense mutation in exon 8 of *F9* was identified in all tested patients. The mutation causes a thymidine-to-guanine transversion at nucleotide position 31094, replacing tyrosine with aspartate at amino acid 371 (Fig 1B). This mutation is a novel variation which has not been reported either in the Hemophilia B Mutation Database (http://www.factorix.org/; Thompson & Chen, 1993) or the Human Gene Mutation Database (HGMD, http://www.hgmd.cf.ac.uk/ac/index.php), although a 31095A>C variation has been reported in HGMD leading to a Y371S mutation causing mild HB (20% of normal FIX activity). As residue 371 is located in the highly conserved serine protease domain (Fig 1C), we decided to generate mouse models containing the

corresponding mutations to further confirm the phenotype as well as to explore the potential of CRISPR/Cas9-mediated genome editing for amelioration of HB in mice.

### Generation and characterization of *F9* mutant mouse strains

We employed the Cas9 system as previously described (Shao *et al*, 2014) to introduce a precise point mutation in the mouse *F9* locus corresponding to human amino acid residue 371 (position 381 in mouse FIX, Fig 1C). After injection of Cas9 components and distinct donor templates into mouse zygotes (Fig 2A), three mouse strains, respectively, bearing the novel mutant site $F9^{Y381D}$ and two previously reported sites ($F9^{Y381S}$ and $F9^{383STOP}$) were generated (Fig 2B and Appendix Fig S1). No off-target cleavage was detected through T7E1 analysis and sequencing of the 10 most likely potential off-target sites (Appendix Fig S2). The *F9* mRNA level in liver tissue was not impaired in $F9^{Y381D}$ and $F9^{Y381S}$ mice but significantly decreased in $F9^{383STOP}$ mice since the premature stop codon dramatically affects mRNA stability (Fig 2C). Consistently, the protein level was almost invisible in $F9^{383STOP}$ mice but not in $F9^{Y381D}$ mice compared to wild-type controls (Fig 2C), suggesting that the Y381D mutation does not influence actual FIX levels and stability. The average aPTT for 8-week-old wild-type mice was $22.04 \pm 0.77$ s ($n = 5$), but aPTTs were prolonged to $41.78 \pm 1.18$ s and $45.684 \pm 1.10$ s for the $F9^{383STOP}$ ($n = 5$) and the $F9^{Y381D}$ ($n = 5$) mouse strains, respectively (Fig 2D). The aPTT for the $F9^{Y381S}$ strain was not significantly increased (aPTT = $22.71 \pm 1.17$ s; $n = 5$; Fig 2D) which was consistent with observations in humans. No significant difference of average prothrombin time (PT) was observed in 8-week-old wild-type, $F9^{Y381S}$, $F9^{Y381D,}$ and $F9^{383STOP}$ mouse strains (Fig 2E). The prolonged aPTT and the normal PT demonstrated that the $F9^{Y381D}$ and $F9^{383STOP}$ mouse strains generated in this study are models for HB. Additionally, these strains were subjected to a tail-clip challenge for further confirmation of defects in hemostasis. The blood loss volume of the 4 strains of mice in 5 min after tail-clip was recorded as shown in Fig 2F. $F9^{383STOP}$ mice ($n = 5$) lost more than 10 times the volume of blood lost by wild-type controls ($n = 5$), but no significant difference was observed between wild-type controls and $F9^{Y381S}$ ($n = 6$) or $F9^{Y381D}$ mice ($n = 13$), suggesting that the Y381D mutation does not cause very severe HB in mice. However, the survival rate of $F9^{383STOP}$ and $F9^{Y381D}$ mice was significantly decreased in the two days after the tail-clip challenge (Fig 2G). Taken together, these data indicate the successful generation of HB mouse models and suggest that the Y381D and 383STOP mutations dramatically disrupted FIX activity.

### Restoration of hemostasis in HB mice through naked DNA injection of Cas9 components

To explore whether correction of a mutated *F9* gene *in situ* can restore clotting activity in adult HB mice, we employed CRISPR/Cas9 system-mediated genome editing. As DNA-based vectors do not have viral contaminants and almost no immunogenicity (Kay, 2011), we first explored hydrodynamic tail vein (HTV) injection, a sophisticated strategy for *in vivo* gene delivery in animals. The pX458 plasmid containing Cas9-2A-GFP components (Fig 3A) was delivered through our modified HTV injection procedure to test

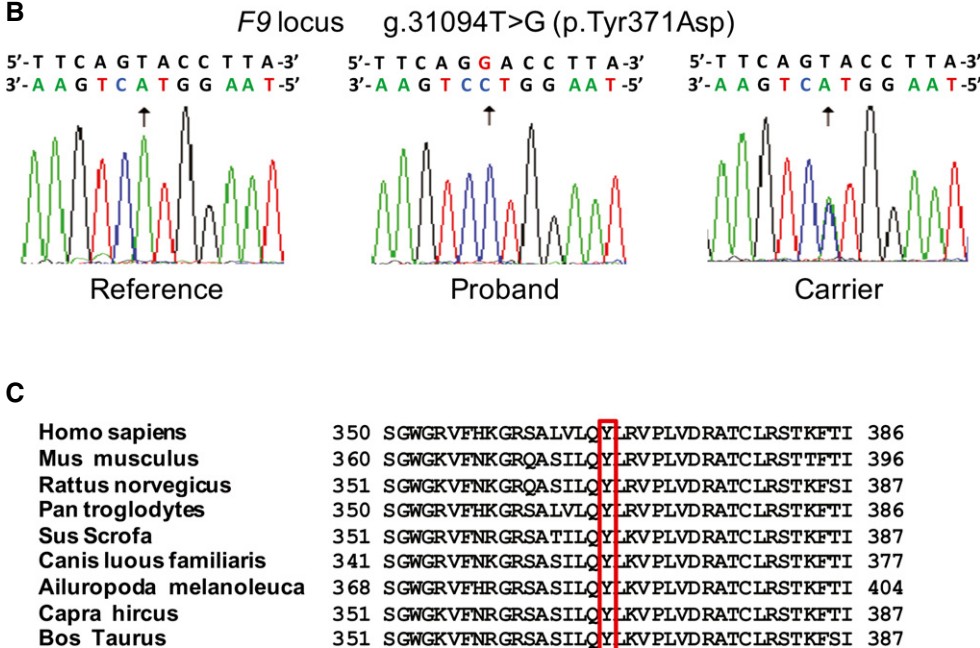

**Figure 1. Characterization of a *F9* gene mutation in a family with a history of bleeding diathesis.**

A   Family pedigree. Blackened symbols indicate the patients with hemophilia B; white circles with dots show carriers of the mutation; open symbols indicate healthy individuals. Circles represent females; squares represent males. The proband is labeled with an arrow.

B   The partial sequences of the *F9* gene from a healthy subject (left), the patient (middle), and a carrier (right).

C   Amino acid alignment of partial sequence of the serine protease domain of FIX in 10 species. An asterisk (*) indicates positions which have a single, fully conserved residue. A colon (:) indicates conservation between groups of strongly similar properties. A period (.) indicates conservation between groups of weakly similar properties.

the transgenic efficiency. Twenty hours after pX458 plasmid injection, liver tissue was obtained and eGFP expression was detected in $18.2 \pm 3.1\%$ of hepatocytes on average (Appendix Fig S3). Donor templates as either a 120-nt ssODN or a plasmid were delivered with pX458 through HTV injection (Fig 3A). To prevent donor DNA cleavage by Cas9/sgRNA, the HDR donor plasmid contains the G>T corrected nucleotide and 10 synonymous single-nucleotide exchanges which are flanked by about 400-bp

homologous arms on each side (Fig 3A and E). The average aPTTs for WT and $F9^{Y381D}$ mutant mice were $22.1 \pm 1.75$ s and $51.88 \pm 2.71$ s, respectively (Fig 3B). Eight weeks after injection of Cas9/ssODN or Cas9/donor plasmids, the average PT was not affected but aPTTs significantly dropped to $32.7 \pm 2.02$ s ($P = 0.0004$ compared to $F9^{Y381D}$) and $31.6 \pm 4.99$ s ($P = 0.0046$ compared to $F9^{Y381D}$), respectively (Fig 3B). Additionally, the aPTT of Cas9/donor plasmid (px458 + donor)-treated $F9^{Y381D}$ mice

**Table 1. Factor IX clotting activity of members in the hemophilia B family.**

| Subject | FIX functional activity (% of normal level) |
|---|---|
| Proband (IV2) | 2 |
| Proband's twin brother (IV3) | 2 |
| Proband's non-twin brother (IV1) | 70 |
| Proband's grandfather (II1) | 3 |
| Proband's aunt (III3) | 55 |

Normal FIX activity range (%): > 45

was significantly shortened compared to the mock-treated (mock + donor) group ($P = 0.0306$), but had no significant difference compared to the WT group ($P = 0.0892$). Similar results were also obtained when we used ssODN as donor templates (Fig 3B). To further confirm the therapeutic effect, tail-clip challenge assay was performed. The survival rate was increased from 38% (5 out of 13) in the untreated group to 86% (12 out of 14) in $F9^{Y381D}$ mice which had received Cas9/donor plasmid injection (Fig 3C). Our data suggest that Cas9-mediated hepatic genome editing corrected the *F9* mutation *in situ* and significantly restored the coagulation activity of adult $F9^{Y381D}$ mice. Sequencing analysis of 177 TA-clones suggested that the indel rate and HDR rate both were 0.56% in the ssODN group (Fig 3D). Through deep sequencing analysis, the modification efficiency was determined in the group that received plasmid donor and Cas9/sgRNA. About 4.39% of *F9* alleles were modified, including 2.84% indel mutations and 1.55% G>T corrections (Fig 3D and E). In three randomly selected individual mice, the highest HDR rate was 2.84% with 2.77% indel mutations (Fig 3F), suggesting that the plasmid donor exhibits a high fidelity of corrective repair. The majority of HDRs resulted in the desired G>T correction and total or partial synonymous substitutions (Fig 3E). These data suggested that the presence of the corrected gene in about 0.56% of endogenous *F9* alleles is sufficient to restore clotting activity. No significant difference in blood aspartate transaminase (AST) and alanine transaminase (ALT) levels between control and naked DNA-injected group was observed at 8 weeks (Fig 3G), and liver tissues were histologically normal despite a mild increase of inflammatory cytokine mRNA levels (Fig 3H), suggesting that

HTV injection of naked DNA vectors was well tolerated in accordance with a previous report (Yin *et al*, 2014).

## Genetic correction of *F9* mutation via Cas9 system delivered through recombinant adenovirus

Next, we sought to increase the transduction efficiency of the Cas9 components to achieve a higher HDR rate in mouse hepatocytes. An adenoviral (Adv) gene delivery system was employed due to its large DNA capacity, high efficiency, and non-integration into the host genome (Crystal, 2014). More importantly, genome editing using a protein-capped AdV was more accurate than using other vectors (Holkers *et al*, 2014). We generated adenoviral Cas9 (AdvCas9) and a vector containing the corrective donor template following a sgRNA target (AdvG/T; Fig 4A). To test the infection and editing efficiency, $1 \times 10^{10}$ and $7 \times 10^{10}$ vector genomes of AdvCas9 and AdvG/T were delivered per mouse via tail vein injection. Four days later, almost all hepatocytes were infected (Fig 4B) and caused an about 19% mutation frequency detected through T7E1 assays (Fig 4C and Appendix Table S5). Additionally, no off-target mutations were detected in hepatocytes (Appendix Fig S4). To our surprise, the aPTTs were not shortened in $F9^{Y381D}$ mice 8 weeks after treatment in both the low-dose group ($1 \times 10^{10}$ and $1 \times 10^{10}$ vector genomes of AdvCas9 and AdvG/T) and the high-dose group ($1 \times 10^{10}$ and $7 \times 10^{10}$ vector genomes of AdvCas9 and AdvG/T) compared with untreated $F9^{Y381D}$ mice (Fig 4D). The indel rate and HDR rate of the low-dose group were both 0.85% according to sequencing of 237 individual TA-clones (Fig 4E). In high-dose group, deep sequence analysis suggested that genetically 31.34% of *F9* alleles were targeted through either NHEJ (25.81%) or the HDR (5.53% G>T correction) pathway (Fig 4E). Further studies demonstrated dramatically increased plasma AST and ALT levels and the transcription of inflammatory factors in the high-dose group (Appendix Fig S5A and B), suggesting that AdVs induced immune response-mediated hepatic injury (Muruve *et al*, 1999), which was confirmed by histology analysis (Appendix Fig S5C). Severe hepatic toxicity was perhaps mainly due to the high immunogenicity of the AdV-induced systemic cytokine response. Although low-dose AdV administration did not stimulate significant inflammation (Appendix Fig S5), the extremely low genome editing efficiency suggested that there is still a long way to go to optimize AdV for curing HB through Cas9-mediated gene correction.

---

**Figure 2. Generation and characterization of variant *F9* mutant mouse strains.**

A  Schematic diagram of the strategy to generate $F9^{Y381D}$ and $F9^{Y381S}$ mouse strains. Introduced mutant oligonucleotide is in red, and the corresponding amino acids are in green.

B  Sanger sequencing showing the point mutations in *F9* mutant strains. Mutated oligonucleotides are indicated by arrows, and the premature stop codon is underlined.

C  Expression of *F9* mRNA and protein in hepatic tissue of indicated mouse strains. Data represent means ± SE. The significant effect was obtained using the two-tailed unpaired Student's *t*-test to determine the *P*-value. M: Marker; arrowhead: FIX protein; Note: the bands of FIX in $F9^{383STOP}$ lanes are faint with small molecular weight smears. The gel is representative of three independent experiments.

D  Measurement of coagulation activity by aPTT in mice at 8 weeks of age. $n = 5$ for each group, data represent means ± SE. *P*-value was determined using two-tailed unpaired Student's *t*-test.

E  Measurement of coagulation activity by PT in mice at 8 weeks of age. $n = 5$ for each group, data represent means ± SE.

F  Measurement of blood loss over a 5-min period after tail transection in mice at 8 weeks of age. Wild-type, $n = 5$; $F9^{Y381S}$, $n = 6$; $F9^{Y381D}$, $n = 13$; $F9^{383STOP}$, $n = 5$. *P*-value was determined using two-tailed unpaired Student's *t*-test.

G  Survival rate of mice after the tail-clip challenge. The mice were monitored for 2 days after tail clipping. Wild-type, $n = 5$; $F9^{Y381S}$, $n = 6$; $F9^{Y381D}$, $n = 13$; $F9^{383STOP}$, $n = 5$.

Source data are available online for this figure.

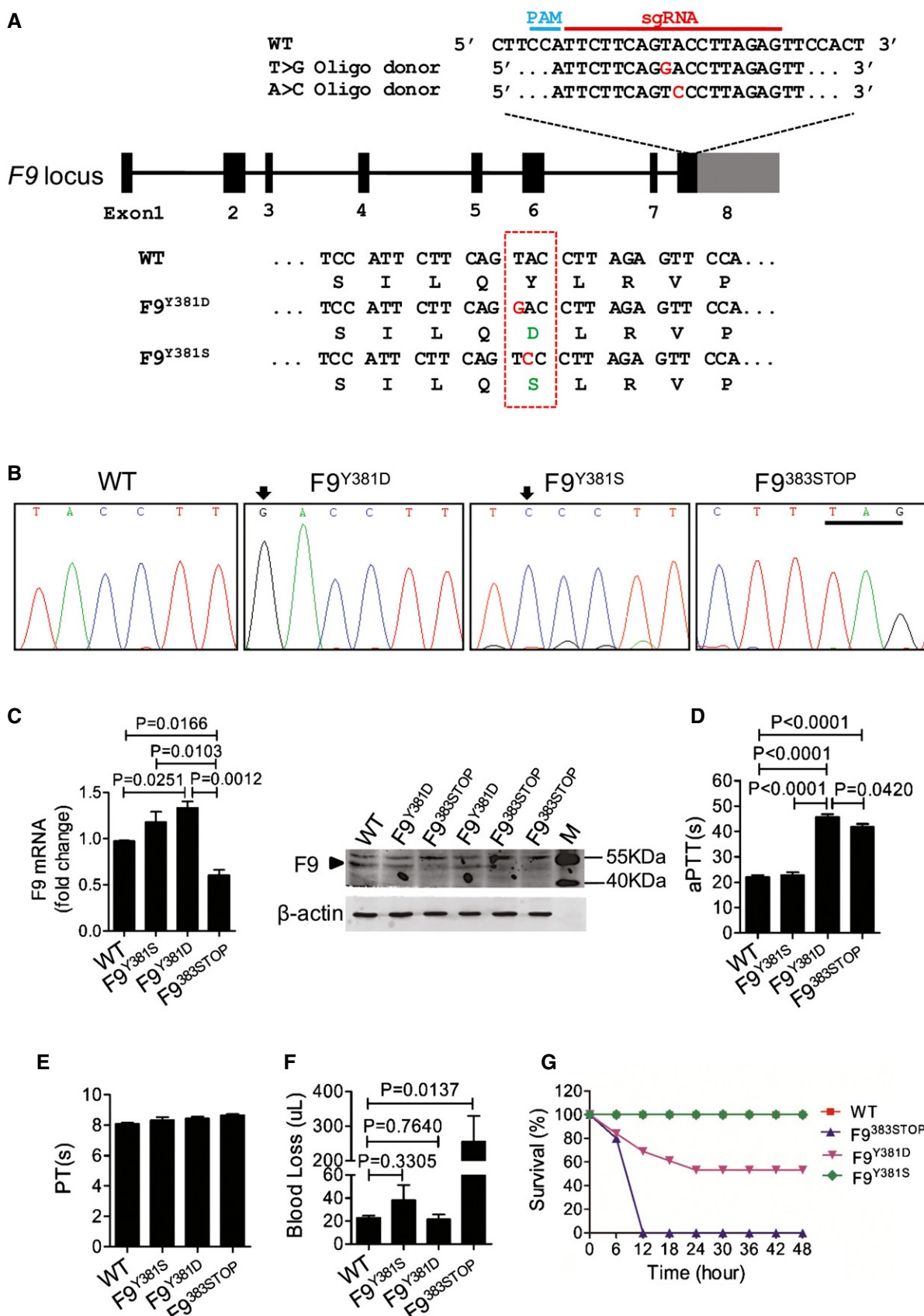

Figure 2.

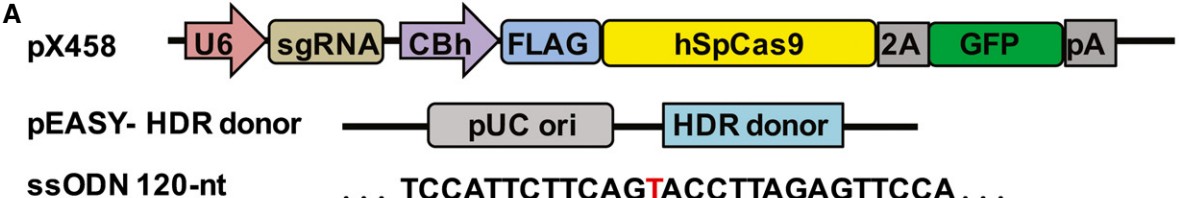

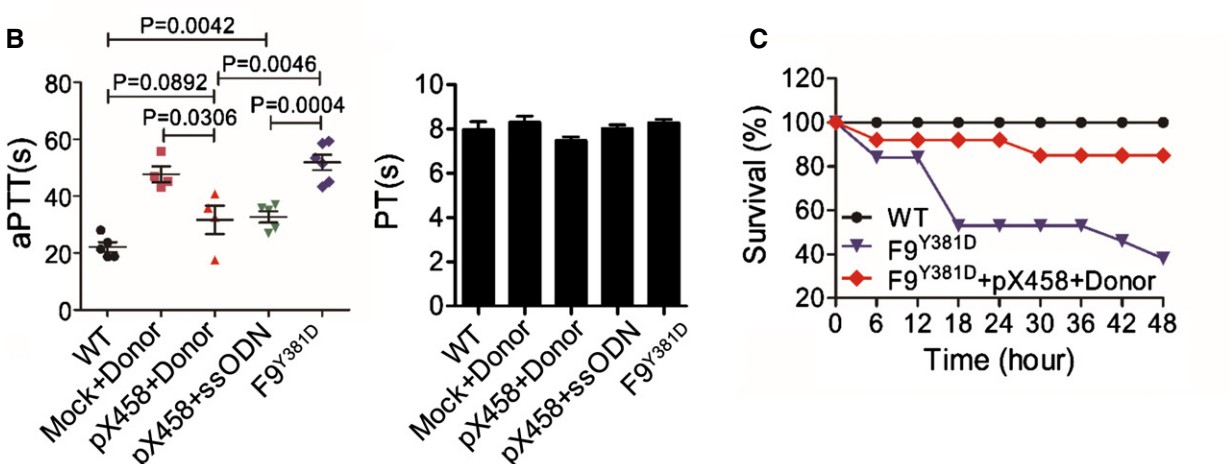

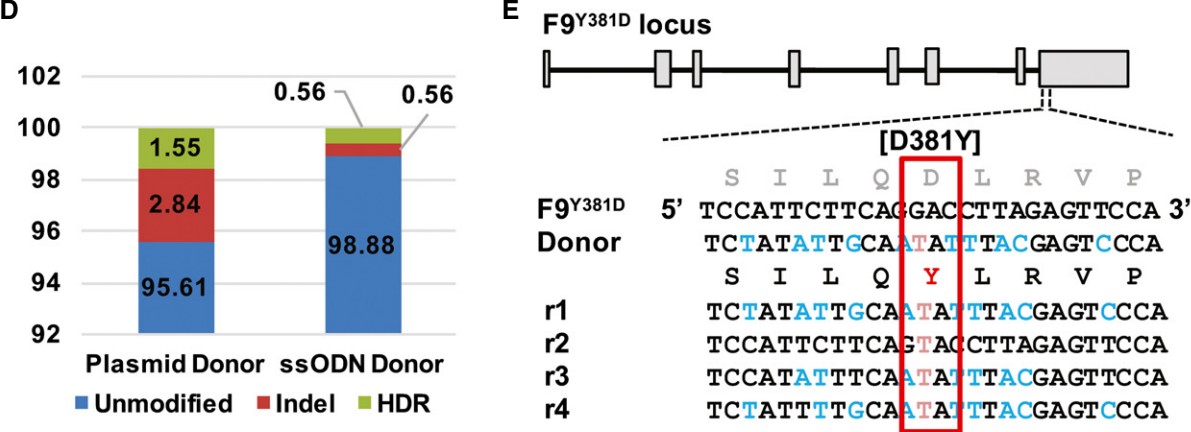

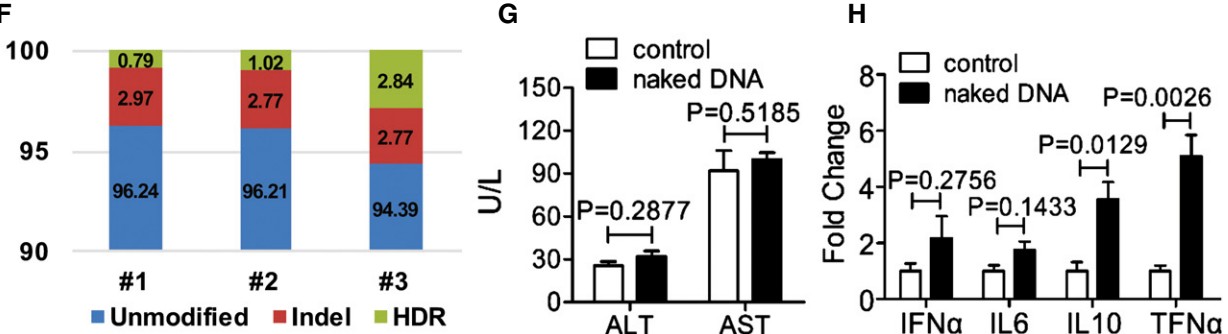

Figure 3.

**Figure 3.  Amelioration of HB in *F9* mutant mice through naked DNA injection of Cas9 components.**

A    Schematic diagram of plasmids used for treatment *in vivo*. Cas9 protein and sgRNA were from pX458 vector. The plasmid donor was 800 bp in length in pEASY vector. The ssODN donor was 120 oligonucleotides.

B    Test of clotting activity by aPTT and PT at 8 weeks following hydrodynamic tail vein injection of 120 μg pX458 and 120 μg donor plasmids (or 120 μg ssODN) per mouse. Data are presented as mean ± SE. The experiment was replicated three times. *P*-value was determined using two-tailed unpaired Student's *t*-test.

C    Survival rate of mice after the tail-clip challenge. $F9^{Y381D}$ mice were treated with or without Cas9/sgRNA/donor DNA for 3 months. After tail-clip challenge, the survival rate of each group over 2 days was determined. Wild-type, $n = 9$; untreated $F9^{Y381D}$, $n = 13$; DNA-treated $F9^{Y381D}$, $n = 14$.

D    Frequency of genetic modification in $F9^{Y381D}$ hepatocytes was determined either by deep sequencing (Cas9/donor vector-treated group) or by TA-clone sequencing (Cas9/ssODN group).

E    HDR donor design for correction of Y381D and representative Illumina sequencing reads ($r_n$) in DNA-treated HB mice. Red text indicates the correction of mutation, whereas blue text indicates the synonymous mutations.

F    The genome editing efficiency of individual mice is presented.

G    Plasma aspartate aminotransferase (AST) and alanine aminotransferase (ALT) level was determined in $F9^{Y381D}$ mice 8 weeks after HTV injection ($n = 5$). Data are presented as mean ± SE. *P*-value was determined using two-tailed unpaired Student's *t*-test.

H    mRNA levels of inflammatory cytokines from liver tissue of naked DNA-treated mice were determined by real-time PCR. Data represent means ± SE. The experiment was replicated three times. *P*-value was determined using two-tailed unpaired Student's *t*-test.

## Discussion

To date, at least 1113 unique *F9* mutations, including 812 unique point mutations, have been identified worldwide (Rallapalli *et al*, 2013). Here, we report a novel Y371D mutation of the *F9* gene and confirmed a more severe HB phenotype than the previously identified Y371S mutation suggesting that this single amino acid substitution dramatically affects FIX activity. To our knowledge, this is the first report to confirm the FIX activities of distinct patient-derived mutations in the mouse. Subsequently, we provide the first evidence demonstrating amelioration of HB in mice through Cas9-mediated genetic correction in the endogenous *F9* locus.

In experiments with HTV injection, the HDR rate was 0.56% in mice injected with the ssODN donor and 1.5% in the group with the double-stranded DNA (dsDNA) plasmid donor. It suggests that correction of 0.5–1.5% of endogenous *F9* alleles in hepatocytes is sufficient to ameliorate hemostasis in HB mice, but the repair efficiency for fully curing HB is not clear. Previous studies suggest that higher efficiency was achieved when using ssODN as templates compared to dsDNA in cell culture (Lin *et al*, 2014). However, in our study, a higher repair efficiency was observed in the group using dsDNA as a donor template probably due to the shorter half-life of ssODN in live animals. The HDR rate through HTV injection of naked DNA in our study is similar to that reported by Yin and colleagues (Yin *et al*, 2014). Although due to the safety issue and rapid extinction, hydrodynamic injection of a DNA vector is not considered to be clinical practicable at the present time, modifications of the procedure for hydrodynamic gene therapy of liver segments have been achieved in pigs and have been demonstrated to be safe in a clinical trial (Khorsandi *et al*, 2008). Since DNA-based vectors offer several advantages over state-of-the-art recombinant viral vectors (Kay, 2011), especially their low host immunogenicity which makes them applicable for repetitive treatments, the clinical potential of CRISPR/Cas9-mediated genome editing via hydrodynamic injection holds considerable promise.

Compared with DNA-based vectors, the non-integrating AdV-mediated gene delivery system significantly increased CRISPR/Cas9 system-mediated genome editing efficiency *in vivo*. After AdV treatment, 5.5% of mutant *F9* alleles were corrected in HB mice, which is much higher than the rate through HTV injection. However, no significant restoration of clotting activity was observed in AdV-treated HB mice in either the high-dose group or the low-dose group. We were surprised that no therapeutic effect was observed in the low-dose AdV-treated group. It is probably due to the relatively low HDR efficiency and an adverse effect of AdV injection on clotting as a previous study suggested that aPTT increased after injection of helper-dependent AdVs even in healthy mice (Brunetti-Pierri *et al*, 2004). It is critical to reduce the side effects of AdV-mediated gene delivery for further studies of CRISPR/Cas9-mediated gene therapy. The high efficiency and non-integrating features of AdV can be used to correct defective cells for *ex vivo* gene therapy, since protein-capped AdV donor DNA greatly reduces illegitimate recombination (Holkers *et al*, 2014). The recombinant adeno-associated virus (rAAV) system is the state-of-the-art gene delivery system which has been demonstrated to be safe and effective in FIX deficiency-induced HB patients (Nathwani *et al*, 2014). Although rAAV-mediated site-specific repair of a mutated *F9* gene has successfully restored hemostasis in HB mice (Li *et al*, 2011; Barzel *et al*, 2015), it will be valuable to evaluate the genome editing efficiency in an HB model using rAAV to deliver a short form of Cas9 derived from

**Figure 4.    Genetic correction of *F9* mutation via the Cas9 system delivered through AdV.**

A    Schematic of AdVs used for *in vivo* genome editing. AdvCas9: humanized spCas9 is fused to mCherry through 2A peptide. AdvG/T: an 800-bp corrective template and sgRNA targeting the mouse *F9* Y381D mutation was inserted into an AdV containing GFP.

B    *In vivo* infection efficiency 4 days after Adv injection. Inserts: negative control from untreated mouse liver tissue. Scale bar: 100 μm.

C    *T7E1* assay was performed on liver DNA obtained from samples in (B), and the frequency of Cas9-induced indels is indicated as "%" below the lane.

D    Measurement of clotting activity by aPTTs and PTs 8 weeks after tail vein injection with indicated vectors. Data represent means ± SE. The experiment was replicated three times. *P*-value was determined using two-tailed unpaired Student's *t*-test.

E    *In vivo* genome editing efficiency of whole liver tissues from mice treated with Adv(Cas9 + G/T). (Left) HDR donor used for correction of Y381D mutation, and Illumina sequencing reads ($r_n$) are presented. Red text indicates the correction of the *F9* mutation, whereas blue text indicates the intended synonymous mutations. (Right) The percentage of indel or HDR corrections obtained through sequencing of 235 individual TA-clones (low dose) or Illumina sequencing (high dose) of treated mouse liver tissues.

Source data are available online for this figure.

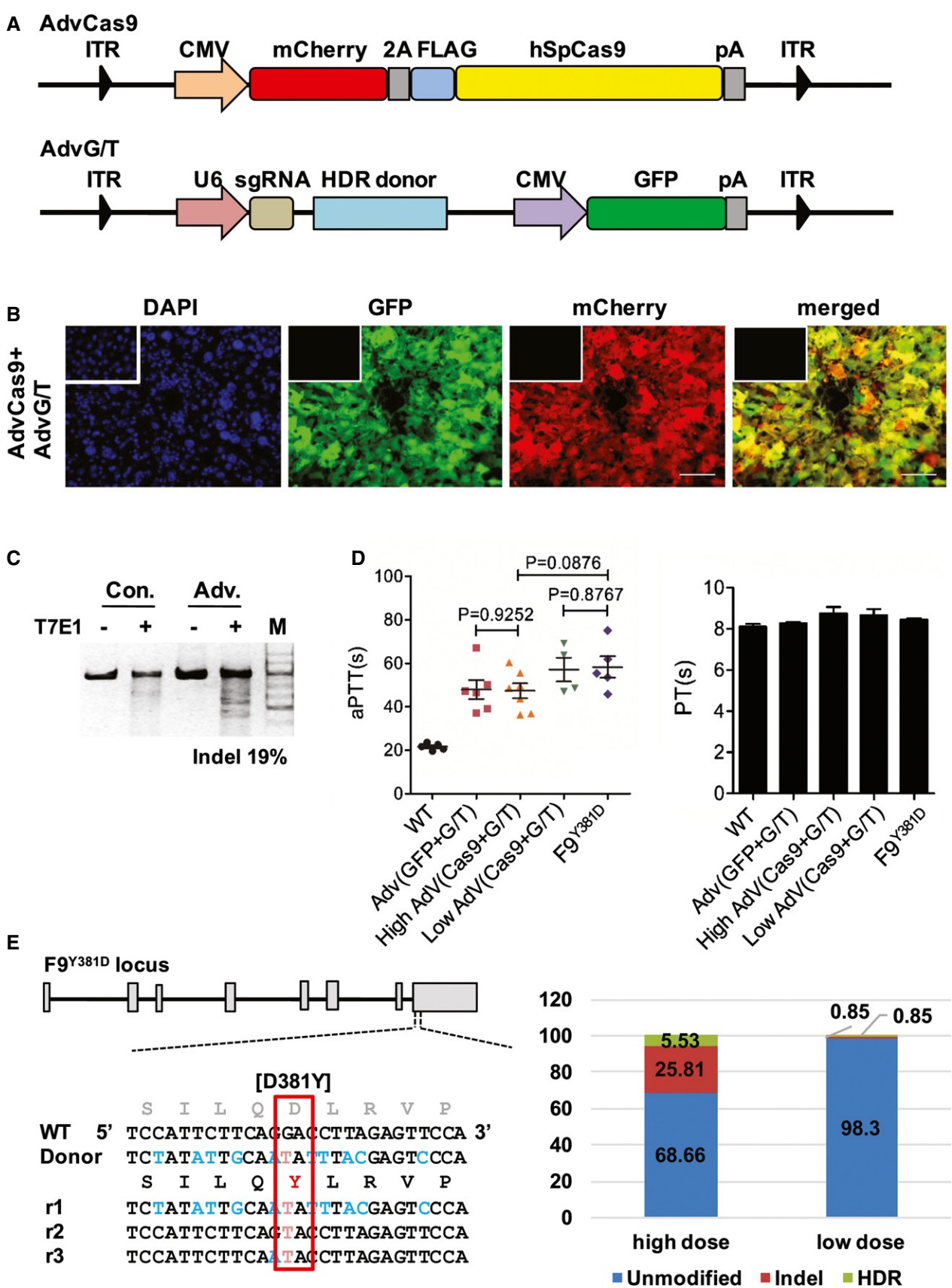

**Figure 4.**

*Staphylococcus aureus* (Ran *et al*, 2015) since CRISPR/Cas9 is the most efficient genome editing tool currently available. In conclusion, we identified a novel *F9* mutation in patients and ameliorated HB in adult mice through *in situ* correction of the *F9* locus via Cas9-mediated genome editing. Our approach provides proof of principle for considering CRISPR/Cas as a versatile tool in precision medicine, from functional confirmation of genomic variants to the development of therapeutic solutions, directed against a broad array of genetic diseases.

# Materials and Methods

### Study approval

All participating individuals were informed and gave documented consent prior to participation. The studies were approved by the Human Ethics Committee of the Affiliated Hospital of Hainan Medical University following the principles set out in the WMA Declaration of Helsinki and the Department of the Health and Human Service Belmont Report. All animal experiments conformed to the regulations drafted by the Association for Assessment and Accreditation of Laboratory Animal Care in Shanghai and were approved by the East China Normal University Center for Animal Research.

### Patients and mutation analysis

The proband presented to the Affiliated Hospital of Hainan Medical University, with repeated spontaneous hemorrhage in his left knee joint, and was diagnosed with hemophilia. Peripheral blood DNA of the proband, his twin brother, his parents, and some available relatives was collected for genetic studies. The entire coding sequence of the *F9* gene was amplified by polymerase chain reaction (PCR) using published primers (Chang *et al*, 2002). The amplified product was purified using an AxyPrep DNA Gel Extraction Kit (Axygen Biosciences, USA) and sequenced using ABI 3500 Genetic Analyzer (Applied Biosystems, Foster City, CA, USA). Sequencing results were analyzed using BLAST (http://www.ncbi.nlm.nih.gov/blast) program against the human normal *F9* gene sequence (GenBank Accession No. K02402), and the mutation was compared with the Human Gene Mutation Database (http://www.biobase-international.com/product/hgmd) and Factor IX Mutation Database (http://www.factorix.org/). Blood DNA was extracted and sequenced from 100 symptom-free residents in the same community as the proband to see whether the *F9* variant is a DNA polymorphism or disease-causing mutation.

### Generation of the mouse hemophilia B model

The mouse hemophilia B model was generated by co-injection of Cas9 mRNA, sgRNA (Appendix Table S1), and ssODNs (Appendix Table S2) as described previously (Shao *et al*, 2014). Briefly, superovulated female C57BL/6J mice were mated to male C57BL/6J mice, and embryos were collected from oviducts. Cas9 mRNA (100 ng/µl), sgRNA targeting the FIX loci (50 ng/µl), and ssODN carrying the Y381D or Y381S (10 ng/µl) mutations were co-injected into the pronuclei of one-cell embryos. The injected embryos were cultured in KSOM overnight before they were

transplanted into pseudopregnant mice. One week after birth, genomic DNA from the toes or tail of the newborn F0 mice was extracted for sequencing. Mice were housed in standard cages in a specific pathogen-free facility on a 12-h light/dark cycle with *ad libitum* access to food and water.

### Adenovirus amplification

Adenoviral Cas9 (AdvCas9) and the corrective adenovirus containing the donor template and the sgRNA (Appendix Table S1) target (AdvG/T) were purchased from Obio Technology (Shanghai). Twenty-four hours before infection, 293A (ATCC) cells were seeded into 15 dishes (15 cm in diameter) and cultured until 80% confluency. For each dish, $5 \times 10^8$ particles of adenovirus were added and mixed well. Seventy-two hours later, 1.5 ml 10% Nonidet P40 (NP40) was added into each plate after removing the media and the cells were dislodged and lysed by pipetting. Cell lysate was centrifuged at 13,800 *g* for 10 min, and the supernatant was collected. For 100 ml supernatant, 50 ml virus precipitation buffer (20% PEG8000, 2.5 M NaCl) was added, followed by 1-h incubation on ice. The virus particles were spun down at 13,800 *g* for 30 min and dissolved in 1.10 g/ml CsCl solution (1.32 g CsCl in 10 ml 20 mM Tris–HCl buffer pH 8.0). A CsCl density gradient was prepared by sequentially dripping down 8 ml 1.40 g/ml CsCl solution, 12 ml 1.3 g/ml CsCl solution, and 20 ml virus containing solution into a Beckman Quick-Seal tube (Item No. 342414). The virus was centrifuged at 46,000 *g* in 4°C for 3 h. A concentrated virus band between 1.3 and 1.4 g/ml CsCl solution was collected and dialyzed (20% sucrose (w/v), 10 mM Tris–HCl and 2 mM MgCl$_2$, pH 8.0) to remove the CsCl.

### Tail vein injection of adenovirus

The injection volume of virus was adjusted to 100 µl with phosphate-buffered saline (PBS), pH 7.4 (Gibco). For the high-dose group, mice (8 to 10 week old) were injected with $1 \times 10^{10}$ and $7 \times 10^{10}$ particles of AdvCas9 and AdvG/T, respectively; for the low-dose group, mice (8 to 10 week old) were randomly grouped and injected with $1 \times 10^{10}$ and $1 \times 10^{10}$ particles of AdvCas9 and AdvG/T. For the pilot experiment, 4 days after injection, mice were sacrificed by carbon dioxide asphyxiation. Whole liver samples were harvested and portioned for immunofluorescence, T7E1 assay, and Sanger sequencing. For the second experiment, 8 weeks after injection, liver samples were split for H&E staining, inflammatory cytokines quantification, and HDR detection. Terminal blood samples were collected for aPTT and the aminotransferase test.

### Plasmids construction and hydrodynamic tail vein injection

pSpCas9(BB)-2A-GFP (pX458) was a gift from Feng Zhang (Addgene plasmid # 48138). The protospacer sequence targeting the Y381D mouse locus was synthesized from Genewiz and inserted into pX458 through BbsI. The HDR donor vector was constructed by insertion of the donor sequence into the pEASY-Blunt Simple vector (TransGen Biotech). ssODNs (120 nt; Appendix Table S2) carrying the D381Y were synthesized from Genewiz. Hydrodynamic injection was performed as described previously (Liu *et al*, 1999) with minor modifications. Briefly, all dosages of plasmid solution were diluted

into 1 ml (final volume) 0.9% NaCl. The tail vein injection was performed within 5–7s for maximum liver absorption. For the plasmid donor group, mice were injected with both pX458 (120 μg) and pEASY-HDR donor (120 μg); for the ssODN donor group, mice were injected with pX458 (120 μg) and ssODN-HDR donor (120 μg). The control group received 1 ml 0.9% NaCl only.

### On-target, off-target, and HDR analysis

Dissected mouse liver tissue (0.1 g) was fast frozen in liquid nitrogen, homogenized, and digested in lysis buffer (400 mM NaCl, 100 mM Tris–HCl (pH 8.0), 5 mM EDTA, 0.2% SDS, 20 g/ml RNase A, and 500 g/ml Proteinase K). Liver genome DNA was isolated by phenol–chloroform extraction. To detect cleavage and HDR efficiency, PCR amplicons of on-target products were subcloned to the pEASY-blunt vector for sequencing. The number of clones for on-target analysis of Cas9/ssODN group and low-dose Adv group was over 100. Off-target prediction was implemented using the CRISPR design website (http://crispr.mit.edu/). The primers for PCR amplification are listed in Appendix Table S3 or Appendix Table S4 as indicated. On-target PCR products of Cas9/donor plasmids group and high-dose Adv group were subject to deep sequencing. Deep sequencing libraries were made from 1 to 100 ng of the PCR products using Nextera protocol (Illumina). Libraries were normalized through quality check and sequenced on Illumina HiSeq 2500/3000 machines (125 bp, paired-end). Every sequencing library produced 10 M reads. Reads were mapped to the PCR amplicons as references. Data processing was performed according to standard Illumina sequencing analysis procedures. The high throughput sequence data from this publication have been submitted to the Sequence Read Archive database, and the accession number is PRJNA299277.

### Immunofluorescence, Western blot, and Real-time PCR

Mouse liver tissues were fixed in 4% paraformaldehyde, dehydrated, embedded in OCT, and cryosectioned at 4 μm thickness. Thereafter, sections were rinsed in PBS to remove the OCT, stained with DAPI, and sealed using mounting buffer (Sigma-Aldrich). Cas9 and sgRNA expression was indicated by mCherry and GFP, respectively, under the microscope. For Western blot, liver tissue was homogenized in liquid nitrogen and the total protein content was released in lysis buffer (50 mM Tris-base (pH 7.4), 150 mM NaCl, 1% Triton X-100, 1% sodium deoxycholate, 0.1% SDS, and Complete Mini Protease Inhibitor cocktail (Roche)). The expression of GFP and FIX protein was visualized by anti-GFP antibody (Santa Cruz) and anti-F9 polyclonal antibody (Protein tech). For qPCR, liver tissue was homogenized in liquid nitrogen and the total RNA was isolated with RNAiso Plus (TaKaRa). RT–PCR was performed with SYBR Green (TaKaRa). The primers for qPCR were listed in Appendix Table S6.

### Aminotransferase test

Mouse plasma was collected by retro-orbital bleeding and stored at 4°C for 30 min. Thereafter, the supernatants of the samples were collected after centrifuging at 860 *g* for 10 min at 4°C. The amount of aminotransferase was measured by the AU680 Chemistry System (Beckman).

### The paper explained

#### Problem

Hemophilia B, an X-linked genetic bleeding disorder caused by deficiency of coagulator factor IX, is a worldwide hematology disease. Whether direct correction of the *F9* mutation by targeting the endogenous locus is sufficient for restoration of clotting activity through somatic genome editing is still not determined.

#### Results

We identify a family with hemophilia B carrying a novel mutation, Y371D, in the human *F9* gene. We used the CRISPR/Cas9 system to generate distinct genetically modified mouse models and confirmed that the novel Y371D mutation resulted in a more severe hemophilia B phenotype than the previously identified Y371S mutation. We subsequently deliver Cas9 components targeting the *F9* Y371D mutation in adult mice. Our results show correction of 0.56% of endogenous *F9* alleles in hepatocytes, which was sufficient to restore hemostasis in hemophilia B mice.

#### Impact

Our studies suggest that CRISPR/Cas9-mediated *in situ* genome editing is a feasible therapeutic strategy for hemophilia B and in general for human hereditary diseases, although an efficient and clinically relevant delivery system must be developed for clinical use.

### Activated Partial Thromboplastin Time (aPTT) and prothrombin time (PT) test

Mouse plasma was obtained via retro-orbital bleeding, diluted 9:1 into sodium citrated anticoagulant, followed by gently mixing. Blood samples were spun at 220 *g* for 15 min, and the upper layer was carefully separated from red blood cells for the aPTT and PT test. The aPTT and PT tests were measured by the CA-1500 automatic blood coagulation analyzer (SYSMEX).

### Tail-clip challenge

Mice were anesthetized, and a tail-clip assay was performed as a previous described (Kung *et al*, 1998; Park *et al*, 2015). In brief, the distal part of the tail at 1.5 mm diameter was cut and allowed to bleed for 5 min freely. Blood samples were collected, and the total bleed volume of each mouse was measured. After holding firm pressure on the tail for 1 min, the mice were monitored for 2 days after clipping and the survival rate of each group was determined.

### Statistics

Data are expressed as means ± SEM. Means of two groups were compared using Student's *t*-test (unpaired, 2-tailed), with $P < 0.05$ considered to be statistically significant.

**Expanded View** for this article is available online.

### Acknowledgements

We thank Dr. Stefan Siwko for scientific editing and comments and Meizhen Liu for microinjection assistance. This work was partially supported by grants from the State Key Development Programs of China (2012CB910400 and 2012CB966502), grants from the National Natural Science Foundation of China

(no. 31371455, 81202104, 81330049, 81060016, 81460034, 81260032, 81202104 and 31140021), International Science & Technology Cooperation Program of China (2014DFA30180), and a grant from the Shanghai Municipal Commission for Science and Technology (14140900300, 15JC1400201).

## Author contributions

YG, ZS, LM, LW, LrW, LZ, YS, YC, WL, KH, and HH performed the mouse studies, NM, YM, QL, YM, YY, and YH analyzed the clinical data, ML and DL designed the experiments, analyzed the data, and wrote the manuscript.

## Conflict of interest

The authors declare that they have no conflict of interest.

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
