## [Review Process File · EMBO Molecular Medicine]

CRISPR/Cas9 mediated somatic correction of a novel coagulator factor IX gene mutation ameliorates hemophilia in mouse

Yuting Guan, Yanlin Ma, Qi Li, Zhenliang Sun, Lie Ma, Lijuan Wu, Liren Wang, Li Zeng1, Yanjiao Shao, Yuting Chen, Ning Ma, Wenqing Lu, Kewen Hu, Honghui Han4, Yanhong Yu, Yuanhua Huang, Mingyao Liu and Dali Li

Corresponding authors: Dali Li, Mingyao Liu (East China Normal University) and Yanlin Ma (Hainan Medical University)

Review timeline:

Submission date:	08 November 2015
Editorial Decision:	09 December 2015
Revision received:	02 February 2016
Editorial Decision:	15 February 2016
Revision received:	17 February 2016
Accepted:	18 February 2016

Transaction Report:

Editor: Roberto Buccione

1st Editorial Decision

09 December 2015

Thank you for the submission of your manuscript to EMBO Molecular Medicine. I am sorry for not having been able to get back to you sooner

In this case, notwithstanding the current interest and timeliness of the topic, we experienced unusual difficulties in securing three willing and appropriate reviewers. As a further delay cannot be justified I have decided to proceed based on the two available consistent evaluations.

Both Reviewers are quite positive on your manuscript although they raise some issues that require your action. I will not dwell into much detail as their comments are detailed. I would like, however, to highlight a few main points.

As you will see the Reviewers agree on a few basic issues, which require your action. These include the request that actual FIX levels, activity and stability be assayed. Both also suggest that a tail-bleeding assay should be performed to measure correction of the phenotypes. I agree that the above needs to be carried out.

Reviewer 1 has additional comments. S/he notes that the genome editing efficiency of the different CRISPR-Cas9-based approaches should be compared using the same technique (deep sequencing of the targeted locus). However, after Reviewer cross-commenting, it was agreed that this will not be

necessary if the above points are dealt with satisfactorily. You will need, however, to tone down statements on comparing single-stranded donor oligonucleotides and dsDNA efficiencies.

In conclusion, while publication of the paper cannot be considered at this stage, we would be pleased to consider a revised submission, with the understanding that the Reviewers' concerns must be addressed as outlined above with additional experimental data where appropriate and that acceptance of the manuscript will entail a second round of review. I will endeavour to reach a final decision as quickly as possible.

Please note that it is EMBO Molecular Medicine policy to allow a single round of revision only and that, therefore, acceptance or rejection of the manuscript will depend on the completeness of your responses included in the next, final version of the manuscript.

As you might know, EMBO Molecular Medicine has a "scooping protection" policy, whereby similar findings that are published by others during review or revision are not a criterion for rejection. However, I do ask you to get in touch with us after three months if you have not completed your revision, to update us on the status. Please also contact us as soon as possible if similar work is published elsewhere.

Please note that EMBO Molecular Medicine now requires a complete author checklist (<http://embomolmed.embopress.org/authorguide#editorial3>) to be submitted with all revised manuscripts. Provision of the author checklist is mandatory at revision stage; The checklist is designed to enhance and standardize reporting of key information in research papers and to support reanalysis and repetition of experiments by the community. The list covers key information for figure panels and captions and focuses on statistics, the reporting of reagents, animal models and human subject-derived data, as well as guidance to optimise data accessibility.

I also suggest that you carefully adhere to our guidelines for publication in your next version, including presentation of statistical analyses and our new requirements for supplemental data (see also below) to speed up the pre-acceptance process.

I look forward to seeing a revised form of your manuscript as soon as possible.

***** Reviewer's comments *****

Referee #1 (Comments on Novelty/Model System):

In this paper, Guan et al. characterized a novel mutation of F9 gene in a patient with Hemophilia B and generated an appropriate mouse model harboring this mutation, reproducing the hemophilic phenotype observed in the patient. Moreover, they tested different genome editing strategies to restore hemostasis in this mouse model of hemophilia. Some experiments should be performed to confirm these findings and to improve the technical quality of the paper as detailed in the "Remarks to be sent to the author".

Referee #1 (Remarks):

In this paper, Guan et al. characterized a novel mutation of F9 gene in a patient with Hemophilia B and generated an appropriate mouse model harboring this mutation, reproducing the hemophilic phenotype observed in the patient. Moreover, they tested different CRISPR-Cas9-based genome editing strategies to restore hemostasis in this mouse model of hemophilia. A plasmid-based platform is shown to be less toxic and more effective compared to the adenoviral delivery of the CRISPR-Cas9 system. Some experiments should be performed to confirm these findings and to improve the technical quality of the paper.

Major comments:

1. The Authors have to better characterize the effects of the novel mutation in terms of protein concentration and stability in plasma. They should also evaluate the potential impairment of FIX secretion and of its protease activity. We believe that these factors contribute to the extent of correction that is necessary to ameliorate the hemophilic phenotype.

2. The authors should evaluate the long-term correction of the hemophilic phenotype and the persistence of the correction after induced liver regeneration (e.g.: after partial hepatectomy).
3. The Authors have to compare the genome editing efficiency of the different CRISPR-Cas9-based approaches using the same technique (preferably deep sequencing of the targeted locus).

Minor comments:

1. The Authors should collect the blood for the aPTT test using tail bleeding. Retro-orbital bleeding may alter the results of this test because of the potential contamination of the blood samples with Tissue Factor.
2. The Authors should also perform PT assay testing different sample dilutions to better evaluate possible differences among mice groups that might go unnoticed with such short clotting times.
3. Did the Authors notice any difference in the extent of liver damage and in the increase of the inflammation markers between the mice receiving a low dose and a high dose of adenoviral vectors?
4. The Authors should comment on the possibility of using a lower dose of adenoviral vectors to restore hemostasis without any liver damage and inflammation response.
5. The Authors should explain better the rationale of the use of ssODN harboring silent mutations (to avoid the re-cutting of the Cas9-gRNA complex, to better detect the genome editing efficiency) and the different frequency of gene conversion observed in Figure 3C.

Referee #2 (Comments on Novelty/Model System):

The current manuscript deals with correction through homologous recombination of a defect in the factor IX gene through CRISPR-CAS9. This is one of the first reports that shows that a disease phenotype can be corrected in vivo through homologous recombination. As such the impact of the current study is very high.

The medical impact is high since this paper provides proof-of-concept for gene therapy through homologous recombination in a small animal model.

Referee #2 (Remarks):

This is a very interesting manuscript that shows phenotypic correction of hemophilia B through homologous recombination.

The approach taken is very novel and highly relevant for curative approaches for mono-genetic disorders. As such the potential impact of this study is very high.

Technically the experiments are well-performed. The molecular genetic analyses are of the highest possible level. The phenotypic correction of the factor IX gene is monitored by the APTT; this is a global coagulation test. Actual levels of factor IX in plasma of the "corrected" mice are not available. Measuring factor IX levels in plasma following gene-correction would be very informative and would provide an independent means to demonstrate that a considerable percentage of the mutated FIX gene has indeed been corrected. An ELISA-based assay that estimates the amount of factor IX protein in plasma would be most appropriate.

Lack of tail-vein bleeding following an incision is commonly used to demonstrate correction of a bleeding phenotype in mouse models of hemophilia. It would be nice if this information was also included in the current manuscript.

Minor comments.

Figure 1 describes a novel mutation in the FIX gene. A large number of mutations in the FIX gene have already been described. This information can also be presented as part of the Supplementary information.

Referee #1 (Remarks):

In this paper, Guan et al. characterized a novel mutation of F9 gene in a patient with Hemophilia B and generated an appropriate mouse model harboring this mutation, reproducing the hemophilic phenotype observed in the patient. Moreover, they tested different CRISPR-Cas9-based genome editing strategies to restore hemostasis in this mouse model of hemophilia. A plasmid-based platform is shown to be less toxic and more effective compared to the adenoviral delivery of the CRISPR-Cas9 system. Some experiments should be performed to confirm these findings and to improve the technical quality of the paper.

Major comments:

Comment 1. *The Authors have to better characterize the effects of the novel mutation in terms of protein concentration and stability in plasma. They should also evaluate the potential impairment of FIX secretion and of its protease activity. We believe that these factors contribute to the extent of correction that is necessary to ameliorate the hemophilic phenotype.*

Response: Thanks for the reviewer's comment. We tried our best to use all commercially available ELISA kits against mouse FIX and 2 kits against human FIX, but none of them can detect secreted mouse FIX. For this reason, we could not determine the actual FIX level in mice. Alternatively, we detected the mRNA and protein level of FIX in mouse hepatic tissue. After testing 3 antibodies claimed to be against mouse FIX, we found one of them worked. Since the novel mutation is a point mutation, we found it did not affect the stability of F9 mRNA or protein but the mutation harboring a premature stop codon greatly impaired F9 mRNA and protein level. These data have been presented in revised Figure 2C. Unfortunately, since the commercial available commercial FIX activity assay kit is also for human FIX, we were unable to evaluate mouse FIX activity. In order to evaluate the extent of correction, we used tail bleeding assays which we explain in detail to answer the comment below.

Comment 2. *The authors should evaluate the long-term correction of the hemophilic phenotype and the persistence of the correction after induced liver regeneration (e.g.: after partial hepatectomy).*

Response: To evaluate the long-term correction effect, we employed a tail-bleeding assay beginning 12 weeks after the mice had received tail vein injection. As shown in revised Figure 3C, it suggested that the correction of F9 mutation ameliorated the HB phenotype for a prolonged period. Although it will be interesting to induce liver regeneration in our model to test the correction, we used naked DNA but not viral system to deliver Cas9/sgRNA and donor template which were very unstable in mice. In addition, the donor template was not a full-length F9 cDNA which could not produce any protein. For these two reasons, we think the therapeutic effect is from corrected endogenous FIX rather than expression of exogenous FIX DNA.

Comment 3. *The Authors have to compare the genome editing efficiency of the different CRISPR-Cas9-based approaches using the same technique (preferably deep sequencing of the targeted locus).*

Response: We used TA-clone sequencing for Adv, ssODN and plasmid treated groups (over 120 clones/mouse for each group). To increase the accuracy, we employed deep sequencing and showed the data for plasmid DNA injected group. We found that the correction efficiency obtained by these two strategies were comparable, so we did not use deep sequencing for the other two groups. The editors also suggested that deep sequencing is not necessary, so we compared the correction efficiency of the two groups either using dsDNA or ssODN as donor templates.

Minor comments:

Comment 4. *The Authors should collect the blood for the aPTT test using tail bleeding. Retro-orbital bleeding may alter the results of this test because of the potential contamination of the blood samples with Tissue Factor.*

Response: We used both aPTT and tail bleeding assays to evaluate the HB phenotype in the revised manuscript as shown in Figures 2G and 3C.

Comment 5. *The Authors should also perform PT assay testing different sample dilutions to better*

evaluate possible differences among mice groups that might go unnoticed with such short clotting times.

Response: Originally, we did perform PT assays when testing aPTT but did not present the data. In revision, we showed the PT assays alone with aPTT.

Comment 6. *Did the Authors notice any difference in the extent of liver damage and in the increase of the inflammation markers between the mice receiving a low dose and a high dose of adenoviral vectors?*

Response: We noticed that the mice receiving a low dose Adv did not exhibit very severe liver damage, in contrast to what was observed in high dose Adv treated mice. These data are presented in supplementary figure S5 in the revised version.

Comment 7. *The Authors should comment on the possibility of using a lower dose of adenoviral vectors to restore hemostasis without any liver damage and inflammation response.*

Response: In revised manuscript, we explore this issue as the reviewer suggested in the *Discussion*.

Comment 8. *The Authors should explain better the rationale of the use of ssODN harboring silent mutations (to avoid the re-cutting of the Cas9-gRNA complex, to better detect the genome editing efficiency) and the different frequency of gene conversion observed in Figure 3C.*

Response: We explained the issue as the reviewer suggested in page 8 in the revised manuscript.

Referee #2 (Remarks):

This is a very interesting manuscript that shows phenotypic correction of hemophilia B through homologous recombination.

The approach taken is very novel and highly relevant for curative approaches for mono-genetic disorders. As such the potential impact of this study is very high.

Comment 1: *Technically the experiments are well-performed. The molecular genetic analyses are of the highest possible level. The phenotypic correction of the factor IX gene is monitored by the APTT; this is a global coagulation test. Actual levels of factor IX in plasma of the "corrected" mice are not available. Measuring factor IX levels in plasma following gene-correction would be very informative and would provide an independent means to demonstrate that a considerable percentage of the mutated FIX gene has indeed been corrected. An ELISA-based assay that estimate the amount of factor IX protein in plasma would be most appropriate.*

Response: As we answered in Comment 1 of reviewer 1, we attempted ELISA-based assays to measure the actual FIX level in mutant mice or corrected mice, but all the kits we tried did not work for mouse. As an alternative approach, we examined the mRNA and protein level of FIX in the mouse liver and employed tail bleeding assays as an independent means to show the hemophilic phenotype.

Comment 2: *Lack of tail-vein bleeding following an incision is commonly used to demonstrate correction of a bleeding phenotype in mouse models of hemophilia. It would be nice if this information was also included in the current manuscript.*

Response: As the reviewer suggested, we performed the tail bleeding assays and the data were presented in revised Figure 2G and Figure 3C.

Minor comments.

Comment 3: *Figure 1 describes a novel mutation in the FIX gene. A large number of mutations in the FIX gene have already been described. This information can also be presented as part of the Supplementary information.*

Response: Thanks for the reviewer's comment. Since there are about one thousand mutations has been recorded, we have cited two main database (Hemophilia B Mutation Database (<http://www.factorix.org/>) or the Human Gene Mutation Database (HGMD, <http://www.hgmd.cf.ac.uk/ac/index.php>)) which will help the readers to find more information. In addition, the manuscript has 5 supplementary figures which is the limitation of EMM. Since the

main topic of the manuscript is Cas9-mediated gene therapy, we think that it is better not to substitute any original supplementary figure.

2nd Editorial Decision

15 February 2016

Thank you for the submission of your revised manuscript to EMBO Molecular Medicine. We have now received the enclosed reports from the reviewers that were asked to re-assess it. As you will see the reviewers are now globally supportive and I am pleased to inform you that we will be able to accept your manuscript pending the following final amendments:

- 1) Please note and respond to Reviewer 2's comment on Fig. 2c concerning the molecular weight of the truncated FIX.
- 2) We are now encouraging the publication of source data, particularly for electrophoretic gels and blots, with the aim of making primary data more accessible and transparent to the reader. Would you be willing to provide a PDF file per figure that contains the original, uncropped and unprocessed scans of all or at least the key gels used in the manuscript? The PDF files should be labeled with the appropriate figure/panel number, and should have molecular weight markers; further annotation may be useful but is not essential. The PDF files will be published online with the article as supplementary "Source Data" files. If you have any questions regarding this just contact me.
- 3) You are welcome to suggest a striking image or visual abstract to illustrate your article. If you do please provide a jpeg file 550 px-wide x 400-px high.
- 4) Please note that we now mandate that all corresponding authors list an ORCID digital identifier. You may do so through our web platform upon submission and the procedure takes <90 seconds to complete. We encourage all authors to supply an ORCID identifier, which will be linked to their name for unambiguous name identification.
- 5) I have introduced some minor changes in the manuscript (Title, Abstract and "The Paper Explained"). Please check and approve (or not) based on the attached manuscript file.
- 6) Please provide 5 keywords in the title page.
- 7) Please remove the appendix figure legends from the manuscript file and place them instead under each figure in the appendix data file.

I look forward to seeing a revised final form of your manuscript as soon as possible.

***** Reviewer's comments *****

Referee #1 (Remarks):

The Authors improved the technical quality of the paper by answering to most of the reviewer's comments. The paper is now suitable for publication.

Referee #2 (Comments on Novelty/Model System):

The technologies used in this manuscript are a very high level. correction in hemophilia B; the approach outlined in this paper provides important novel information on the application of CRISPR/Cas9 system for correction of monogenetic disorders. Therefore the medical impact is

high.

Referee #2 (Remarks):

The revised manuscript has been improved. Gene correction of the F9 Y371S mutation is achieved for 0.56% of the F9 alleles. This results in shortening of the APTT (Figure 3B) and increased survival following tail-clip (Figure 3B). The amount of FIX activity and antigen in plasma has not been measured. Despite the clear hemostatic effect as measured by APTT and tail vein bleeding, it is not clear how much FIX is present in plasma of mice receiving Cas9/donor-plasmids.

Additional comment: the immunoblot displayed in Figure 2C shows expression of F9-Y381D and F9-383Stop. The mouse FIX protein is 421 amino acids long. Introduction of a stop-codon at amino acid position 383 is expected to result in a truncated FIX which is expected to migrate at a lower apparent molecular weight when compared to wild type FIX. This is not observed in Figure 2C. Please provide an explanation for the unexpected size of F9383Stop; please add a molecular weight marker to the Figure. What is the evidence that the indicated band corresponds to FIX? Inclusion of a sample of a hemophilia B (lacking FIX) mice would be highly informative.

2nd Revision - authors' response

17 February 2016

Referee #2 (Remarks):

Comment 1: *The revised manuscript has been improved. Gene correction of the F9 Y371S mutation is achieved for 0.56% of the F9 alleles. This results in shortening of the APTT (Figure 3B) and increased survival following tail-clip (Figure 3B). The amount of FIX activity and antigen in plasma has not been measured. Despite the clear hemostatic effect as measured by APTT and tail vein bleeding, it is not clear how much FIX is present in plasma of mice receiving Cas9/donor-plasmids.*

Response: As the Y381S mutation did not prolong the aPTT, we did not do any gene correction experiments on this strain of mice. Since the Elisa kit and FIX activity kit are not suitable for measurement of mouse FIX level, we could not know the exact amount of FIX in plasma of mice receiving Cas9/donor-plasmids.

Comment 2: *Additional comment: the immunoblot displayed in Figure 2C shows expression of F9-Y381D and F9-383Stop. The mouse FIX protein is 421 amino acids long. Introduction of a stop-codon at amino acid position 383 is expected to result in a truncated FIX which is expected to migrate at a lower apparent molecular weight when compared to wild type FIX. This is not observed in Figure 2C. Please provide an explanation for the unexpected size of F9383Stop; please add a molecular weight marker to the Figure. What is the evidence that the indicated band corresponds to FIX? Inclusion of a sample of a hemophilia B (lacking FIX) mice would be highly informative.*

Response: The reviewer raised a very good point. In our WB image of figure 2c, the bands of FIX in F9^{383STOP} lanes are in very low condense with small molecular weight smears. We think that the faint small molecular weight bands could be the truncated FIX in F9^{383STOP} strain. Since the premature stop codon greatly affect mRNA stability and protein translation, the band of FIX in F9^{383STOP} strain is almost invisible. In the revision, we added a note in the figure legend of Figure 2C for explanation of the issue. We included the molecular maker and labeled the molecular weight of the blot in the revision. It is a very good suggestion to use hemophilia B (lacking FIX) mice as a control, but it is not available for us at this moment.

We sincerely hope that these revisions will satisfy the requirements requested by the reviewers, and thank you for consideration to publish our story in *EMBO Molecular Medicine*.

Corresponding Author Name: Dali Li

Manuscript Number: EMM-2015-06039